# Meta-Learning for Fast Model Recommendation in Unsupervised Multivariate Time Series Anomaly Detection

Jose Manuel Navarro[1]  Alexis Huet[2]  Dario Rossi[3]

[1, 2, 3]Huawei Technologies Co., Ltd.

**Abstract**  Unsupervised model recommendation for anomaly detection is a recent discipline for which there is no existing work that focuses on multivariate time series data. This paper studies that problem under real-world restrictions, most notably: (i) a limited time to issue a recommendation, which renders existing methods based around the testing of a large pool of models unusable; (ii) the need for generalization to previously unseen data sources, which is seldom factored in the experimental evaluation. We turn to meta-learning and propose Orthus, the first meta-recommender for anomaly detection in literature that we especially analyze in the context of multivariate times series. We conduct our experiments using 94 public datasets from 4 different data sources. Our ablation study testifies that our meta-recommender achieves a higher performance than the current state of the art, including in difficult scenarios in which data similarity is minimal: Orthus is able to recommend a model in the top 10% (13%) of the algorithmic pool for known (unseen) sources of data.

## 1 Introduction

Unsurprisingly, there is no single anomaly detection (AD) algorithm that is optimal for every realistic situation. Thus, the task of automatic model selection (i.e., correctly choosing which algorithm and hyper-parameters to use, jointly known as a *configuration* or a *model*) is crucial when working on new data. In a supervised learning setting, in which there is available ground truth, this problem has been extensively studied (see He et al. [2021], Karmaker et al. [2021] and references therein). However, model selection becomes especially difficult in an unsupervised setting, given the lack of an objective method for comparing and ranking different algorithmic configurations.

This paper investigates meta-learners for automating anomaly detection on multivariate time series. We consider constraints of a practical scenario, in which model recommendation needs to be as fast as possible and the recommender might work with new datasets from known sources (e.g., sources it has been fed at training time) or with previously unseen and radically different sources (so that generalization ability is particularly important). In particular, we design Orthus, a two-headed meta-recommender, that attains performance superior to the state of the art not only for known sources, but especially for radically different sources: our ablation study shows that (i) a meta-recommender is needed to disambiguate the cases of known/unknown data sources, (ii) a hierarchical organization of the recommenders through clustering improves performance for the known sources case and (iii) the use of well-known time series features (i.e., Lubba et al. [2019]) is both faster and more accurate than current ad hoc features.

We evaluate our contributions with the largest set of real data ever tested on this discipline to date, combining four different public data sources, for a total of 94 datasets. On this benchmark, Orthus selects models in the top 10% of all tested configurations for known data, and is able to internally adapt to novel data sources with limited performance loss (top 13%). We open source the software library we developed[1] to carry out our experiments, including pre-computed performance and meta-feature matrices as well as our recommenders, with the objective of easing reproducibility and fostering research in the area.

---

[1]https://doi.org/10.6084/m9.figshare.22320367

Table 1: Comparison of meta-learning methods in literature.

| Work | Method | Data used | Code |
|------|--------|-----------|------|
| MetaOD | Performance estimation from meta-features | Synthetic groups, independent, non temporal | ✓ |
| ELECT | Nearest Neighbor in performance space | Synthetic groups, independent, non temporal | |
| LOTUS | Nearest Neighbor in Gromov-Wasserstein space | Independent, non temporal | |
| *Orthus (ours)* | *Adaptive performance estimation from meta-features* | *Real grouped, multivariate, temporal* | ✓ |

## 2 State of the art

Meta-learning for Algorithm Selection is a family of methods based on the mapping between algorithmic performance and any information that can be extracted in an unsupervised fashion from datasets, called meta-features, which allows it to be applied on a new dataset (Vanschoren [2018]). These require a large computational effort beforehand in order to train the model recommender, but tend to have a small computational footprint in application and are thus fit for our settings. While meta-learning works are varied for areas as supervised learning (Dyrmishi et al. [2019]) and neural networks (Hospedales et al. [2021]), the problem has only recently started being tackled for unsupervised anomaly detection, as summarized in Table 1.

MetaOD by Zhao et al. [2021] is the first meta-learning scheme in literature directly tailored to unsupervised outlier detection. It is based on collaborative filtering: $C$ configurations are tested over $N$ different datasets, and a matrix factorization process approximates the performance of all configurations based on a projected matrix of meta-features extracted from the datasets. A multivariate random forest regressor is trained to transform the original projected meta-features matrix into the optimized one. Given a new dataset, $X_{new}$, its meta-features are extracted, projected and transformed by the random forest regressor, which allows it to be multiplied by the other matrix factorization component, yielding a performance prediction for every configuration. With respect to MetaOD, Orthus (i) includes a meta-recommender to explicitly account for the novelty of $X_{new}$ (see Section 3.3.1), (ii) differs in the design of the mapping methods (iterative factorization versus singular value decomposition), including the optimization of the normalized discounted cumulative gain versus the mean root squared error, which allows us to perform more complex recommendations (Section 3.3.3), and finally (iii) on the addition and use of time-series specific meta-features (Section 3.4).

A second branch of approaches includes ELECT (Zhao et al. [2022]) and LOTUS (Singh and Vanschoren [2022]). The former is based on establishing distances in the performance space and finding the closest neighbors to $X_{new}$. As new datasets are unlabelled, ELECT estimates performance with a subset of the unsupervised metrics compiled by Ma et al. [2021]. LOTUS searches for nearest neighbors according to the Gromov-Wasserstein distance, for which it proposes a low rank approximation. Additionally, as direct comparison is not possible since code for ELECT and LOTUS is not available, we implement a nearest neighbor baseline as a proxy of ELECT/LOTUS, that we include in our experimental study. In summary, none of the existing methods directly address time series in neither their considered algorithmic space nor their used methods or meta-features, nor is the importance of novelty detection and differing methods for novel and known points analyzed, which can lead to overfitting on patterns that do not generalize across datasets.

## 3 Unsupervised model recommendation through meta-learning

This section expands the basic details of model recommendation through meta-learning and outlines our novel proposals. Every recommender we present will follow the same scheme, divided in an offline training phase, in which the mapper between unsupervised dataset meta-features is trained, and an online application phase, in which a new dataset is issued an appropriate model recommendation.

## 3.1 Offline training

Starting from (i) an available collection of datasets $D = \{X_1, X_2, ..., X_n\}$ for which ground truth labels about anomalies are available and (ii) a collection of algorithmic configurations $C$, this collection is applied and evaluated over $D$ using a suitable metric. This yields $P$, a performance matrix of dimensions $n \times |C|$. In parallel, a set of meta-features, $M$ is extracted for all $X_i$ in $D$, a meta-feature being any numeric value that's extracted from a whole dataset for which no labels are needed. While a comprehensive set of meta-features is presented by Vanschoren [2018], the first set custom tailored to unsupervised outlier detection was presented in Zhao et al. [2021], which we will call $M_{\text{MetaOD}}$. The only task left in this phase is to build an estimator, $f$, to predict the values in $P$ based on $M$, defined as $\hat{P}$:

$$\hat{P} = f(M). \tag{1}$$

## 3.2 Online application

Once a new dataset $X_{\text{new}}$ is received, for which no ground truth labels are available, this phase's objective is to issue a configuration recommendation. Extracting the same meta-feature set as employed in the offline training, a meta-feature vector $M_{\text{new}}$ is obtained, through which a performance estimation $\hat{P}_{\text{new}} \in \mathbb{R}^{1 \times |C|}$ is obtained as $f(M_{\text{new}})$. As this vector contains the performance estimation of all the configurations in $C$, issuing a model recommendation entails only the selection of the one with maximum estimated performance:

$$c_{opt} = \arg\max \hat{P}_{\text{new}}. \tag{2}$$

## 3.3 Orthus design

Our proposed meta-recommender, Orthus, is composed of three components: a novelty detection mechanism and two model recommenders, specialized to each of the two possible situations (novel / known data). Thus, during fitting, Orthus's novelty lies in its model recommenders, whereas during recommendation, its innovative component is its ability to dynamically select which model recommender to use. We describe each step in detail in the following subsections.

### 3.3.1 Meta-recommendation based on data novelty.
An important component of deploying a model recommender in real-world scenarios setting is its ability to not only perform well with datasets similar to the ones seen during fitting, but especially its ability to also perform well with novel and radically dissimilar datasets. Recognizing this need, a novel contribution in Orthus is to propose a mechanism for novelty detection, wich is used to create a meta-recommender that detects whether datasets are novel or not, and then can dispatch different recommenders to be applied in each case. The process is simple: during application, new data points are projected using Uniform Manifold Approximation and Projection (UMAP), by McInnes et al. [2018], in the meta-feature space and clustered alongside the previous datasets in $M$. The decision of what recommender to used can be taken based on whether the new points belong to clusters with points seen during fitting or not using any off-the-shelf clustering algorithm, like DBSCAN. This usage of the meta-feature space is similar to the one presented by Kadioglu et al. [2010], but we use it to perform meta-recommendation instead of default recommendation. While the UMAP projected space does not preserve distances between points, in this step we only use it to judge whether a new point lies close to any other pre-existing point or not, which is still represented in the projection.

### 3.3.2 Recommender for novel data: URegression (UReg).
In case data radically differs from the one seen at training, Orthus next uses Singular Value Decomposition ($P \approx UDV$) and trains a multi-regressor Random Forest to predict $U$ from $M$, $\hat{U}(M)$:

$$\hat{P} = f_{\text{UReg}}(M) := \hat{U}(M)DV. \tag{3}$$

Internally, the Random Forest is optimizing the mean squared error of the solution, so the estimations obtained by $f_{\text{UReg}}$ can guide the end user in understanding not only which model is recommended but also its expected performance. As the number of singular vectors calculated in the SVD, $k$, is a user defined parameter, this method complexity can be tuned by reducing or increasing the number of Random Forests to train. Additionally, it can be expected that for novel data a highly summarized description (i.e., a low number of components) can be enough to guide model recommendation, as it forces the SVD to extract only the broadest, most general patterns across datasets, which we expect to hold true for novel data.

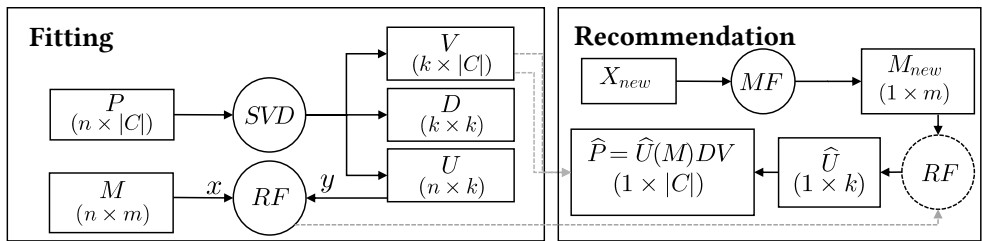

Figure 1: URegression's diagram: $SVD$ represents Singular Value Decomposition, $RF$ a Random Forest multi-regressor model and $MF$ the meta-feature extraction process.

### 3.3.3 Recommender for known data: Clustered Factorization (CFact).

In case data is similar to the one seen during training, Orthus uses a more refined recommender. Given the inherent variety in the algorithmic space we employ, it is expected to have subgroups of AD configurations with similar characteristics, which will be reflected in dissimilar patterns between meta-features and different groups of columns of the performance matrix. As URegression summarizes the whole $P$ together, it can be expected to find more coarse-grained patterns w.r.t. if we could split $P$ in different groups of configurations with similar performance profiles. *CFact* leverages this intuition by performing a UMAP projection of $P^T$ to $d$ (with $d \ll |C|$) and a subsequent clustering of the configurations in the projected space. A *UReg* recommender is fit for each cluster, corresponding to subsets of columns of $P$, obtaining a set of recommenders, $R = \{r_1, r_2, ..., r_{cl}\}$, where $cl$ is equal to the number of determined clusters. As the output of each recommender can be expressed as a configuration and its predicted performance pair $(c_i, p_i)$, when a new dataset is received *CFact* will produce $cl$ tuples of the aforementioned form, with the recommended option being the one that maximizes $p_i$. While the UMAP projected space does not preserve distances between points, it is not an issue in this step as, in the worst case, it will only split datasets in more clusters than needed – however within each cluster the relationship between meta-features and performance remains similar.

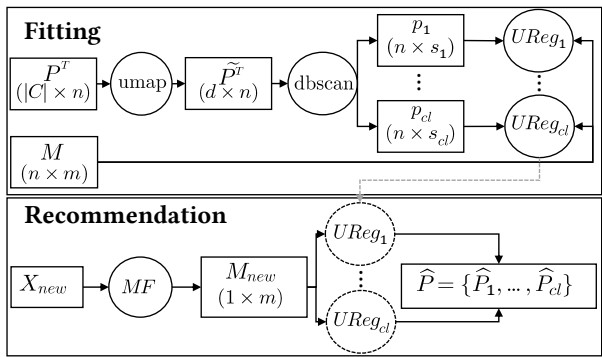

Figure 2: CFact's diagram: *UReg* represents individual URegression models, *umap* a dimensionality reduction of the transposed performance matrix and *dbscan* a clustering over the projected space. This process splits the performance matrix in configuration groups with a similar performance profile, which are fed to specific URegression recommenders.

### 3.4 Time-series meta-features (Catch22)

Finally, we observe that current work Zhao et al. [2021, 2022], Singh and Vanschoren [2022] employs generic meta-features for dataset summarization. For instance, $M_{\text{MetaOD}}$ uses a set of 200 features, some of which requires the application of four anomaly detection methods already (HBOS, Isolation Forest, LODA and PCA) and are as such computationally intensive; additionally, such meta-features do not properly account for the temporal dimension of the data. We therefore set to investigate if, for time-series meta-recommendation, meta-features that are naturally used in the time-series domain provide a competitive advantage: we turn to *Catch22*, a set of 22 univariate time series meta-features. We point out that while these meta-features represent a good starting point (as they have been extracted out of a set of over 4000 for their effectiveness on time-series tasks, are not correlated with each other and are fast to compute Henderson and Fulcher [2021]), other sets could be used in practice (cfr Sec.6). As *Catch22* is extracted from a single series, we define $M_{\text{C22}}$ as the 110 meta-features created by summarizing each of the set of values for each meta-feature by its *{minimum, first quartile, average value, third quartile, maximum}*. This is, for each time series in a dataset, we calculate its 22 meta-features. Then, per meta-feature, we calculate its distribution and extract the summary values previously described. This yields 110 values, which we use as $M$ in our recommenders.

## 4 Methodology

Our objective in the rest of the paper is to compare the performance of anomaly detection model recommenders for time series data, that we collect from quite heterogeneous publicly available sources (Section. 4.1). On these datasets, we train model recommenders using a representative selection of anomaly detection algorithms, spanning over 500 individual algorithmic configurations (Section. 4.2). As for recommenders, we contrast Orthus to the current state of the art, and additionally include relevant methods as a baseline (Section. 4.3). In particular our evaluation aims at assessing the generalization abilities of the recommender to unseen data source, for which we devise two evaluation scenarios (Section. 4.4).

To address the above questions, we designed a software framework that we open source to simplify reproducibility of this paper. For convenience, the software package includes pre-computed $M$ and $P$ matrices and every recommender analyzed in this paper.

### 4.1 Data

Table 2: Data sources summary. Number of rows, columns and anomaly duration are expressed in median ± median absolute deviation.

| Name | Ref. | Datasets | Rows | Columns | Anomaly Length (samples) |
|---|---|---|---|---|---|
| BGP | Putina et al. [2018] | 39 | 1437 ± 31 | 214 ± 70 | 54 ± 1 |
| Benchmark | Lai et al. [2021] | 19 | 400 ± 0 | 5 ± 0 | 1 ± 0 |
| SMD | Su et al. [2019] | 28 | 47405 ± 37 | 31 ± 1 | 11 ± 11 |
| Water | Shin et al. [2020] | 8 | 113401 ± 81393 | 58 ± 3 | 258 ± 159 |
| **Total** | | 94 | 3264992 | 10262 | 92768 |

We used four public data sources of multivariate, numeric time series. They are (i) Server Machine Dataset (SMD), by Su et al. [2019], containing datasets of server metrics, like CPU, memory or network usage; (ii) BGP, by Putina et al. [2018], datasets representing devices deployed on a testbed on which BGP anomalies were manually injected and the features are related to the network control and data planes; (iii) Benchmark, by Lai et al. [2021], a collection of synthetic and realistic datasets for time series outlier detection; and (iv) Water, by Shin et al. [2020], a collection

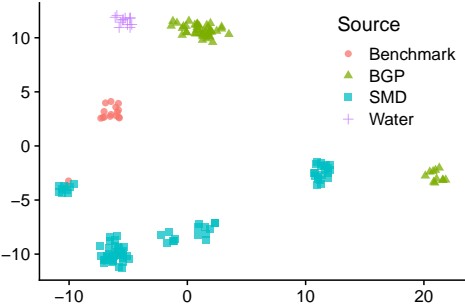

Figure 3: UMAP projection of the datasets in the $M_{Catch22}$ space. Benchmark and SMD are the only data sources with apparent overlap, testifying generality of the considered datasets. Large intra-cluster similarity and inter-cluster dissimilarity are expected when performing model recommendation in an industrial application.

of sensor features from a water-treatment plant devices. Table 2 presents a summary of the number of datasets per source, their size in rows and columns as well as the duration of the anomalies on each of them. To further detail their variety, Fig. 3 displays the 2D UMAP projection of the datasets in the meta-feature space, showing that there is barely no overlap in similarity between sources and that datasets within sources are similar between each other, which is the scenario we expect in practice and that makes generalization abilities especially important.

## 4.2 Anomaly detection algorithms in the Performance Matrix

Regarding anomaly detection algorithms, we aimed for completeness in terms of different algorithmic families. Additionally, in order to capture the potential time series specific anomalies we previously mentioned, we created $P$ with a combination of six among *batch* and *streaming* anomaly detection algorithms: (i) the batch case, we employed Local Outlier Factor (Breunig et al. [2000]), Isolation Forest (Liu et al. [2008]), Random Histogram Forest (Putina et al. [2020]) and LODA (Pevnỳ [2016]), (ii) whereas for streaming algorithms, we opted for Half-Space Trees (Tan et al. [2011]) and xStream (Manzoor et al. [2018]). As for their hyper-parameters, we tested a total of 507 combinations, to ensure a good coverage of their performance space, detailed further on Appendix B.

Each algorithm's output is evaluated using the Area Under the Precision Recall Curve (PRAUC), a common metric for anomaly detection. Then, we project each PRAUC value in a 1-100 scale, as the percentile it represents in the precomputed performance distribution for its dataset, to ensure all values are comparable across datasets. For instance, if the AD algorithms performance distribution for a single dataset were $\{0.2, 0.3, 0.7, 0.75, 0.9\}$, and the recommender selected a configuration with performance equal to 0.75, it would be transformed into 80, as it is equal or larger than 80% of the possible performance values.

## 4.3 Recommenders

In terms of recommenders, we consider **Orthus** (as well as serveral variations in the ablation study where we also consider each individual building block, such as **UReg** and **CFact**) and contrast it to the current state of the art represented by **MetaOD**, by Zhao et al. [2021]. We consider several additional baselines: **Global Best**, which selects the configuration with best median performance in the training data; **Nearest Neighbor (kNN)** by Nikolić et al. [2013], which recommends the model that worked best in the dataset most similar in the meta-feature space to $X_{new}$ and a **Regression** method which builds Random Forest regression models between $P$ and $M$ directly, by Xu et al. [2012]. The parameters used for their comparison are summarized in Table 3.

## 4.4 Experiments

We define two scenarios in terms of the data splitting strategy employed. In *Scenario I*, we perform stratified repeated 10-folds cross validation with 3 repetitions, where each recommender is evaluated a total of 30 times with a 90/10 train/test split, ensuring that datasets from every source

Table 3: Hyper-parameter set used to test the recommenders in the experiments.

| Recommender | Hyperparameters | Recommender | Hyperparameters |
|---|---|---|---|
| Orthus | SVD factors: UReg = 2, CFact = 5
UMAP components = 10
DBSCAN $\epsilon = 1$ | MetaOD | {PCA factors, iterations} = 40, 10
{Min, max} learning rate = 1.05, 0.1
Learning rate steps = 10 |
| kNN | Neighbors k = 1 | Regression | Number of trees = 500 |

are present on each fold: this scenario is the one typically considered in the literature, where therefore, as per Fig. 3, we can expect recommenders to be able to readily leverage information from previous datasets. This constitutes an optimistic scenario in which the recommender has been trained to also include data from the source where it is applied to.

Additionally, we consider a ***Scenario II***, in which we split the data in a leave-one-source-out fashion, creating 4 folds, ensuring in each of them that each data source is *not* present (this creates folds of differing sizes, as each source has a different number of datasets). For each experiment, we fit every recommender with the training data in a single fold and predict the anomaly detection model for corresponding test data of the left-out source. In this case, models have been exposed to a variety of training data, which can be expected however to be quite different from the data where the recommender is applied to. This corresponds to conditions we can expect in realistic scenarios, in which new datasets are not necessarily extracted from the same process or distribution as seen ones. This scenario directly deals with the generalization capabilities of the recommenders.

We repeat each experiment five times with different random seeds, so that presented results aggregated these different repetitions, treating them as different studies in a meta-analysis as per Lipsey and Wilson [2001]. For all reported metrics of interest, we measure their effect size by pooling over the repetitions, estimating its average and its 95% confidence interval (CI).

## 5 Results

We preliminary restrain the set of meta-features (Section 5.1), based on which we report an exhaustive comparison of the recommender performance (Section 5.2). We conclude with an ablation study of our proposal (Section 5.3) and some consideration on computational requirements (Section 5.4).

### 5.1 Preliminary selection of meta-feature sets

To compare the two meta-feature sets we measure, for each recommender, the proportions of cases in which the better performance are obtained using $M_{\mathrm{MetaOD}}$ or $M_{\mathrm{C22}}$. To be conservative, we consider that recommender performance are different only whenever for a dataset $d$ in a scenario $s$ and recommender $r$, the performance difference between the two meta-features set exceeds $|p_{\mathrm{MetaOD}} - p_{\mathrm{C22}}| > \sigma_{d,s,r}^+$, where $\sigma_{d,s,r}^+$ is the maximum of the standard deviation of the performance obtained across seeds for $\{d, s, r\}$ for $M_{\mathrm{MetaOD}}$ and $M_{\mathrm{C22}}$. With this conservative definition, we capture only macroscopic differences and account for the potential variability in recommender behavior across seeds and dataset difficulty.

| | Recommender | | | |
|---|---|---|---|---|
| *Meta-features* | **Orthus** | **MetaOD** | **kNN** | **Regression** |
| *Equal* | 72 ± 2 | 65 ± 3 | 44 ± 2.5 | 68 ± 2.5 |
| $M_{C22}$ | 16 ± 2 | 19 ± 2.5 | 31 ± 2.5 | 18 ± 2 |
| $M_{MetaOD}$ | 12 ± 2 | 16 ± 2.5 | 25 ± 2.5 | 14 ± 2 |

Table 4: Percentage of tests in which the performance of methods selected by each recommender is equal, better for $M_{MetaOD}$ or $M_{C22}$. Estimates and 95% confidence interval across five different random seeds.

Results are reported on Table 4. Briefly, we gather that for all recommenders, performance are similar for the majority of the cases with either $M_{\text{C22}}$ and $M_{\text{MetaOD}}$ meta-feature sets. Additionally, it appears that $M_{\text{C22}}$ is slightly better than $M_{\text{MetaOD}}$, except for the MetaOD recommender itself. Finally, we observe that $M_{\text{C22}}$ features are almost 4× faster to compute with respect to than $M_{\text{MetaOD}}$. In what follows, we limited report results using $M_{\text{C22}}$ and point out that qualitative considerations reported hereafter also holds when using $M_{\text{MetaOD}}$ (detailed results available in Appendix D).

## 5.2 Comparison of recommender performance

We consider all recommenders at a glance, on both mild (I) and hard (II) generalization scenarios, and display their results on Fig. 4 (with alternative finer-grained analysis on Appendix E). Considering the stratified cross validation scenario (I) first, we observe that most approaches appear to have close performance: notice that Orthus recommends on average a model close to the top-10% (89.8). Also, while the differences are statistically significant, we observe that Orthus is only marginally improving over a simple regression (88.7) or global best (84.5) baselines, or over the Meta-OD state of the art (83.7). This confirms previous findings in Zhao et al. [2021] that were indeed showing a naïve solution to provide satisfactory results. Thus, under mild generalization settings, there would be no obvious reasons to suggest for more complex recommender than simple baselines.

At the same time, a completely different picture emerges from the hard generalization scenario (II): in particular, we see in this case that Global best (64.3) and kNN (52.5) approaches, along with state of the art MetaOD (59.7), incur a dramatic performance drop. In this case, simply recommending what worked best in the datasets observed at training time is not useful for the novel data. While direct regression (75.0) is in this case able to interpolate a better recommmendation (at the cost of building one random forest per each of the 507 algorithmic configurations), it is still far lower than Orthus (86.9), whose performance is only minimally affected and thus is able to meet its generalization goal with a top-13% recommendation also in harder settings. We now dissect the fundamental reasons of this performance difference.

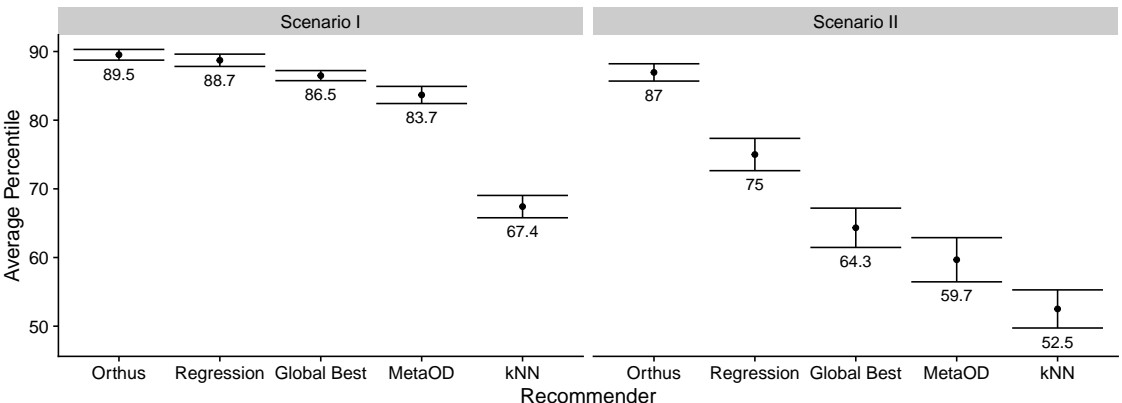

Figure 4: *Recommender comparison*: Average performance percentile obtained by each recommender across five random seeds, expressed as average estimate and 95% confidence intervals.

## 5.3 Ablation study

### 5.3.1 Novel vs known data.
Among the fundamental reasons of the early observed performance difference is the novelty of the data: i.e., recommender strategies that work well with when the recommender is exposed to a sample of the data where it is expected to actuate, no longer apply when the actuation scenario changes drastically. We illustrate this by showing a sensitivity analysis of a building block of Orthus: specifically, URegression internally uses a Singular Value Decomposition (SVD) of $P$, where the number of SVD factors is a free hyper-parameter.

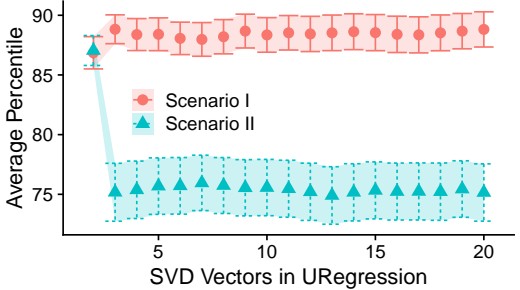

Figure 5: *Ablation study*: UReg performance as a function of the number of SVD factors for Scenario I (stratified cross validation) and II (leave-one-source-out cross validation). Average and 95%CI performance using 5 different random seeds.

Intuitively, for a sufficiently high number of factors, URegression would be equivalent to the Regression recommender we utilize for baseline: hence, as early observed for the Regression recommender, we expect that for growing number of SVD factors the performance of UReg should diverge in the Scenario I and Scenario II. In particular, in the stratified cross validation, Scenario I, we expect performance (and computation time) to increase with the number of SVD factors, while the same does not necessarily hold for the leave-one-source-out Scenario II.

Figure 5 shows the average percentile of UReg for varying number of SVD factors, with 95% CI computed across five random seeds. In practice, (i) we observe a sharp difference between the use of 2 SVD factors and a relatively flat behavior otherwise; (ii) as expected, performance in the stratified validation case improves with the number of SVD factors; (iii) also, we gather that a larger number of factors than 2 is counter-productive for leave-one-source-out scenario II; (iv) finally, it is worth stressing that UReg with 2 SVD factors seems to capture some fundamental dataset properties, with recommendations that are structurally different, and significantly better with respect to those of a global baseline.

These results suggest the existence of global patterns that are present across data sources and are captured correctly just by using few SVD factors, whereas finer-grained relationships captured by using a larger number of vectors don't generalize across sources. While this seems a reasonable explanation, further analysis on a broader set of data sources is needed to confirm the generality of the phenomenon to an even broader set of sources than those considered here.

### 5.3.2 Performance of individual building blocks.
As the main idea in Orthus is, based on the meta-feature projection, to select a recommender that is better suited to handle novel (UReg) or known (CFact) data, we finally contrast Orthus with simpler alternatives where only some of its individual building blocks are used. We fit the internal recommenders to the same scenarios as Orthus and tabulate their average performance percentile with 95% CI and report a rank test for the individual building blocks versus Orthus in Figure 6. The picture confirms that, as expected (i) the CFact recommender is the best choice for Scenario I, where CFact clustering allows for a finer-tuned performance prediction in known data; (ii) the UReg(2) recommender with 2 factors is the best choice for Scenario II, as it captures global patterns that are applicable across unseen sources; finally (iii) Orthus is able to seamlessly meta-recommend the correct one among CFact/UReg(2) based on its novelty estimation capability. Overall, all building blocks are therefore necessary to get accurate recommendations in cases where training and test data are similar, as well as robust recommendations that are able to go beyond recommending a simple default configuration when dealing with radically different data.

### 5.4 Computational requirements

Orthus' requirements are fairly lightweight. On one core of an Intel Xeon E5-4627 CPU, the cost for meta-feature summarization per column is less than 20 ms for $M_{C22}$ (60 for $M_{MetaOD}$) on a 100 row dataset. UReg fitting costs are in the range of 100ms (500ms) for 2 (20) SVD factors, and UReg recommendation costs 25ms (200ms) for 2 (20) factors. The dominant cost in Orthus is represented

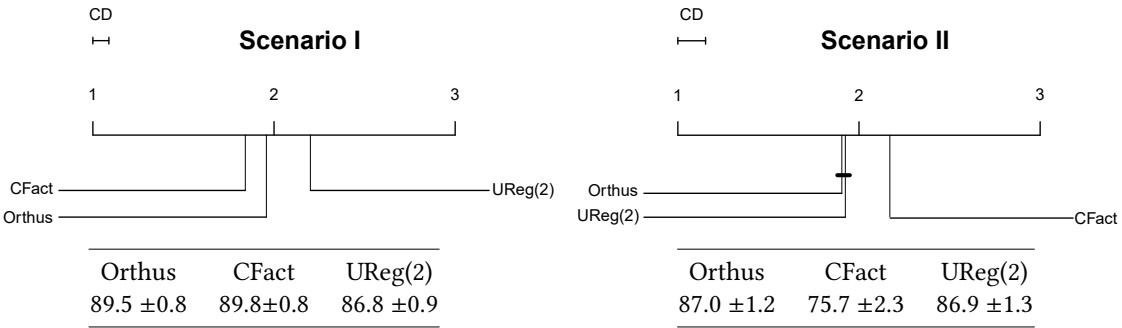

Figure 6: *Ablation study*: Average rank comparison and average performance percentile obtained by Orthus and its individual building blocks, with 95% confidence interval. CD equals the critical distance in average ranking at which recommenders can be considered significantly different.

by the UMAP projection and clustering of the points when a new dataset is received, which takes about 5 seconds in our experiments. We remark, though, that Orthus' fitting (recommendation) times are lower than (equal to) the second recommender in terms of performance, the Regression baseline. Overall, the largest cost during training is the one-time cost incurred by evaluating the 507 combinations of algorithms across all datasets, which take several hours in our setup. Further details on fitting and recommendation times are given on Appendix H.

# 6 Conclusion

In this paper we introduce Orthus, (i) a meta-recommender able to combine two new base recommenders, (ii) URegression and (iii) CFact –that leverage respectively a Singular Value Decomposition of the performance matrix and a clustering of it– and that are designed to provide (ii) general recommendation for novel data, as well as (iii) more accurate recommendation for known data. Our results show that Orthus is able to recommend model configurations close to the top 10% of available models both in mild and hard generalization scenarios.

At the same time, many interesting research questions remain open. First, while Orthus provides better generalization capabilities than the state of the art, a broader study on a much larger set of datasets would be necessary (which are unfortunately not publicly available). Second, while we have shown that time series meta-features (e.g. Catch22) is a valid substitute for custom meta-features set tailored for outlier detection, the meta-feature design space is far from being fully explored, regarding the meta-features themselves (Christ et al. [2018], Fulcher and Jones [2017], Barandas et al. [2020]) as well as the summarization functions used (i.e., higher moments or quantiles). Finally, while we explored recent meta-learning approaches for anomaly detection, Orthus' task could also be classified as an Algorithm Selection problem (Bischl et al. [2016]), which would allow us to compare against solutions in this space. We must also highlight aspects of the present work that deserve future attention, notably concerning the quality of benchmark datasets (i.e., labeling quality and the impact of incorrect labels (as per Wu and Keogh [2021]) as well as limits of standard evaluation metrics (such as PRAUC used as our optimization target, as per Huet et al. [2022]). Regarding Orthus' societal impacts, we have determined that this work presents no notable negative impacts to society or the environment.

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

## A Asset Licenses

MetaOD is released under the BSD-2-Clause license.

# B Hyper-parameters of anomaly detection algorithms

We present in Table 5 the complete grid of hyper-parameter values we tested for each anomaly detection algorithm as well as their source.

Table 5: Hyper-parameter combinations for AD algorithms in the performance matrix $P$ used in our experiments.

| Algorithm (N) | Hyperparameters | Source |
|---|---|---|
| Half-Space Tree (56*) | Number of trees $\in \{5, 10, 50, 100, 200\}$
Tree depth $\in \{5, 10, 20\}$
Initial samples $\in \{1\%, 5\%, 10\%, 20\%\}$ | Code repository |
| Isolation Forest (60) | Sample size $\in \{1\%, 5\%, 10\%, 20\%\}$
Maximum depth $\in \{5, 10, 20\}$
Number of trees $\in \{5, 10, 50, 100, 200\}$ | Isotree |
| LODA (25) | Number of bins $\in \{5, 10, 50, 100, 200\}$
Number of random cuts $\in \{5, 20, 50, 100, 200\}$ | PyOD |
| LOF (36) | Minkowski distance exponent $\in \{1, 2\}$
Leaf size $\in \{5, 10, 20\}$
Number of neighbors $\in \{2, 5, 10, 50, 100, 500\}$ | Scikitlearn |
| RHF (30) | Number of trees $\in \{5, 10, 50, 100, 200\}$
Maximum height $\in \{5, 10, 20\}$
Check duplicates $\in \{\text{True}, \text{False}\}$ | Code repository |
| xStream (300) | Projection size $\in \{5, 20, 50, 100, 200\}$
Number of chains $\in \{5, 10, 50, 100, 200\}$
Chain depth $\in \{5, 10, 20\}$
Initial samples $\in \{1\%, 5\%, 10\%, 20\%\}$ | Code repository |

*4 configurations of HST generated errors during computation and were discarded.

# C Catch22 meta-features

For completeness, we reproduce here the list of the 22 meta-features in Catch22:

1. Mode of z-scored distribution (5-bin histogram).

2. Mode of z-scored distribution (10-bin histogram).

3. Longest period of consecutive values above the mean.

4. Time intervals between successive extreme events above the mean.

5. Time intervals between successive extreme events below the mean.

6. First 1/e crossing of autocorrelation function.

7. First minimum of autocorrelation function.

8. Total power in lowest fifth of frequencies in the Fourier power spectrum.

9. Centroid of the Fourier power spectrum.

10. Mean error from a rolling 3-sample mean forecasting.

11. Time-reversibility statistic.

12. Automutual information, $m = 2, \tau = 5$.

13. First minimum of the automutual information function.

14. Proportion of successive differences exceeding $0.04\sigma$.

15. Longest period of successive incremental decreases.

16. Shannon entropy of two successive letters in equiprobable 3-letter symbolization.

17. Change in correlation length after iterative differencing.

18. Exponential fit to successive distances in 2-d embedding space.

19. Proportion of slower timescale fluctuations that scale with DFA (50% sampling).

20. Proportion of slower timescale fluctuations that scale with linearly rescaled range fits.

21. Trace of covariance of transition matrix between symbols in 3-letter alphabet.

22. Periodicity measure of Wang et al. [2007].

## D Recommender comparison and ablation study using MetaOD's meta-features

This section replicates the experiments displayed in Figures 4 and 6 with $M_{MetaOD}$ instead of $M_{C22}$ as the meta-features extracted from datasets.

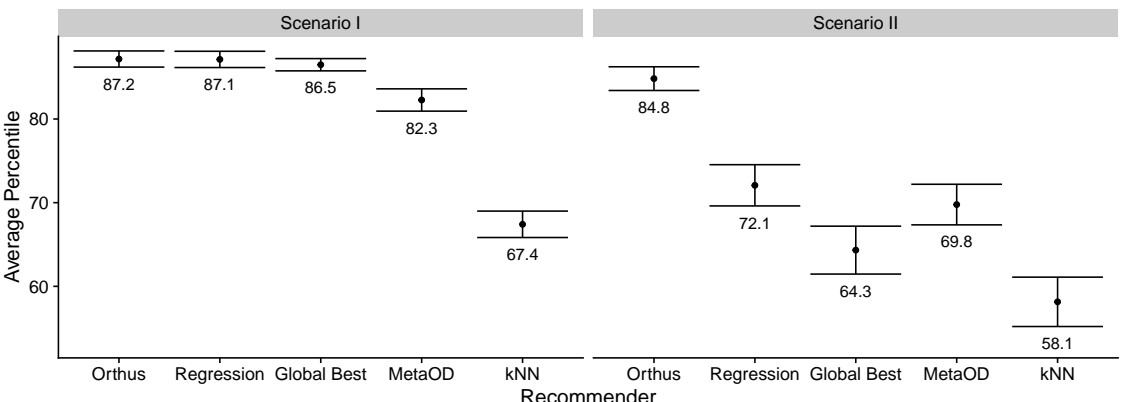

Figure 7: *Recommender comparison using $M_{MetaOD}$*: Average performance percentile obtained by each recommender across five random seeds, expressed as average estimate and 95% confidence intervals.

## E Wilcoxon signed rank test of paired recommenders

As a complement to our analysis in Section 5.2, we display below the p-values obtained by performing Wilcoxon signed rank tests between every recommender across scenarios and the five different random seeds. Recommenders for which the p-value is less than 0.05 can be assumed to perform differently in the tested scenario and seed. Results show that Orthus is significantly different from every recommender across both scenarios except for Regression in Scenario I, where it cannot be assumed to be different. This is due to the small distance on average performance we observed on Fig. 4.

| Recommender 1 | Recommender 2 | Seed | Scenario I | Scenario II |
|---|---|---|---|---|

| | | | | |
|---|---|---|---|---|
| Global Best | Orthus | 1 | 0.000 | 0.000 |
| Global Best | Orthus | 2 | 0.000 | 0.000 |
| Global Best | Orthus | 3 | 0.000 | 0.000 |
| Global Best | Orthus | 4 | 0.000 | 0.000 |
| Global Best | Orthus | 5 | 0.000 | 0.000 |
| Global Best | MetaOD | 1 | 0.245 | 0.000 |
| Global Best | MetaOD | 2 | 0.931 | 0.000 |
| Global Best | MetaOD | 3 | 0.816 | 0.000 |
| Global Best | MetaOD | 4 | 0.487 | 0.845 |
| Global Best | Nearest Neighbor | 1 | 0.000 | 0.000 |
| Global Best | Nearest Neighbor | 2 | 0.000 | 0.000 |
| Global Best | Nearest Neighbor | 3 | 0.000 | 0.000 |
| Global Best | Nearest Neighbor | 4 | 0.000 | 0.000 |
| Global Best | Nearest Neighbor | 5 | 0.000 | 0.000 |
| Global Best | Regression | 1 | 0.000 | 0.001 |
| Global Best | Regression | 2 | 0.000 | 0.000 |
| Global Best | Regression | 3 | 0.000 | 0.000 |
| Global Best | Regression | 4 | 0.000 | 0.000 |
| Global Best | Regression | 5 | 0.000 | 0.000 |
| Orthus | MetaOD | 1 | 0.000 | 0.000 |
| Orthus | MetaOD | 2 | 0.000 | 0.000 |
| Orthus | MetaOD | 3 | 0.000 | 0.000 |
| Orthus | MetaOD | 4 | 0.000 | 0.000 |
| Orthus | Nearest Neighbor | 1 | 0.000 | 0.000 |
| Orthus | Nearest Neighbor | 2 | 0.000 | 0.000 |
| Orthus | Nearest Neighbor | 3 | 0.000 | 0.000 |
| Orthus | Nearest Neighbor | 4 | 0.000 | 0.000 |
| Orthus | Nearest Neighbor | 5 | 0.000 | 0.000 |
| Orthus | Regression | 1 | 0.114 | 0.000 |
| Orthus | Regression | 2 | 0.476 | 0.000 |
| Orthus | Regression | 3 | 0.419 | 0.000 |
| Orthus | Regression | 4 | 0.638 | 0.000 |
| Orthus | Regression | 5 | 0.798 | 0.000 |
| Metaod | Nearest Neighbor | 1 | 0.000 | 0.214 |
| Metaod | Nearest Neighbor | 2 | 0.000 | 0.584 |
| Metaod | Nearest Neighbor | 3 | 0.000 | 0.085 |
| Metaod | Nearest Neighbor | 4 | 0.000 | 0.000 |
| Metaod | Regression | 1 | 0.000 | 0.000 |
| Metaod | Regression | 2 | 0.000 | 0.000 |
| Metaod | Regression | 3 | 0.000 | 0.000 |
| Metaod | Regression | 4 | 0.000 | 0.000 |
| Nearest Neighbor | Regression | 1 | 0.000 | 0.000 |
| Nearest Neighbor | Regression | 2 | 0.000 | 0.000 |
| Nearest Neighbor | Regression | 3 | 0.000 | 0.000 |
| Nearest Neighbor | Regression | 4 | 0.000 | 0.000 |
| Nearest Neighbor | Regression | 5 | 0.000 | 0.000 |

## F  CFact clustering analysis

We report in Fig.9 the number of clusters found by CFact, as well as the distribution of cluster sizes, for every data split in Scenario I, with the internal DBSCAN of CFact configured with $\epsilon = 5$ and minpts = 5. Results confirm that, across splits, there are approximately 10 consistent clusters divided in groups of three different sizes, separating the algorithm families used in the performance matrix.

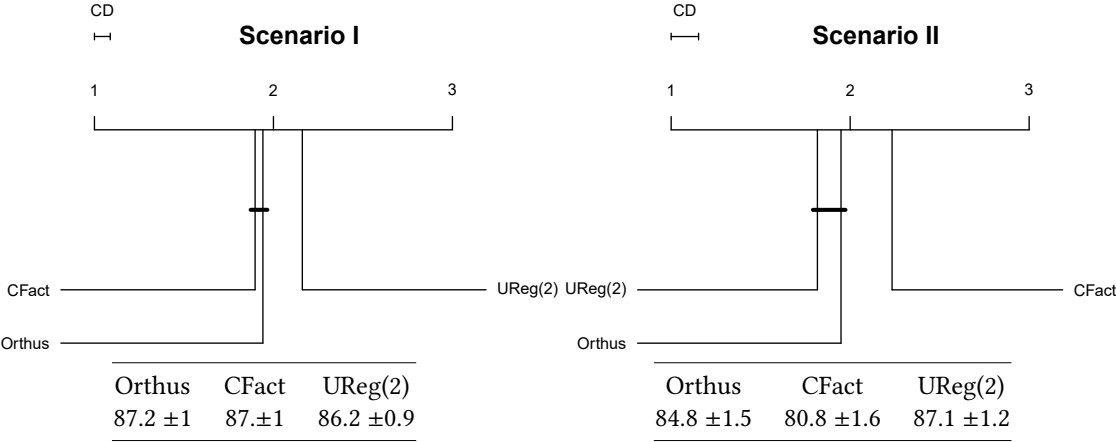

Figure 8: *Ablation study using $M_{MetaOD}$*: Average rank comparison and average performance percentile obtained by Orthus and its individual building blocks, with 95% confidence interval. CD equals the critical distance in average ranking at which recommenders can be considered significantly different.

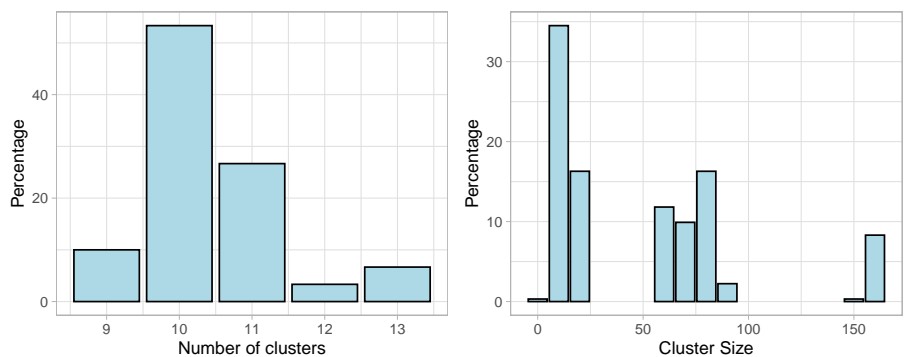

Figure 9: Distribution of number of clusters and cluster sizes found by CFact across the 30 folds of Scenario I.

## G  Ablation study of Orthus' clustering step

To study the influence of DBSCAN's hyper-parameters on Orthus' performance, we display on Fig. 10 the minimum and maximum average performance obtained per parameter value, tested on a single random seed in Scenario I, with $\epsilon \in \{0.1, 0.2, 0.5, 1, 2, 5\}$ and minpts $\in \{2, 5, 10, 20\}$. Results show that, while Orthus' performance is dependent on these values, even extremely low values do not cause the recommender to perform poorly, which indicates the space projected by UMAP that needs to be clustered is an easy problem. This makes Orthus robust to small variations in the clustering parameters. Nevertheless, other clustering algorithms like HDBSCAN could be used transparently with Orthus to reduce its parameter-induced variability.

## H  Computation time and resources used

Preparing the performance matrix $P$ (evaluating 511 configurations over 94 datasets) was completed during a month in several stages in parallel computation on a server with an Intel Xeon E5-4627 CPU with 16 physical cores and 500 GB of RAM.

Regarding computation time of the different recommenders, we tested their fitting and recommendation times for one of the folds of Scenario I (83 datasets for fitting and 11 for recommendation),

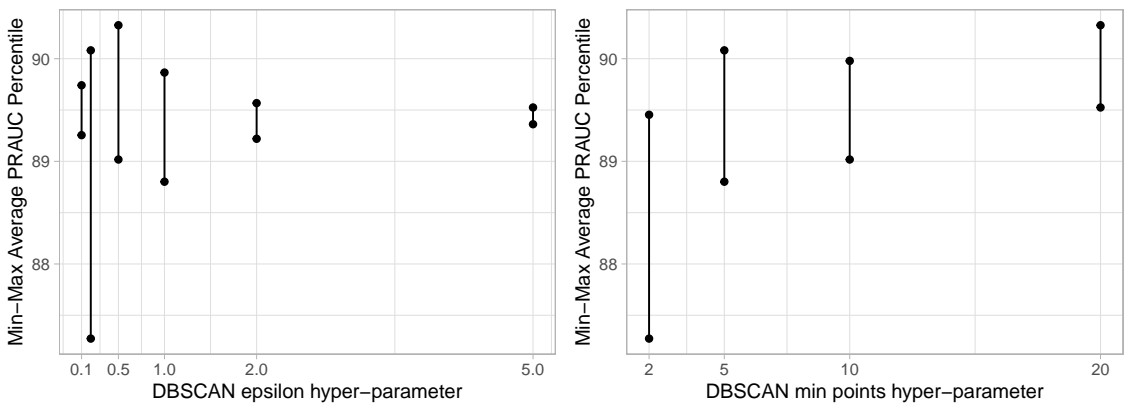

Figure 10: Effect of DBSCAN's hyper-parameters on Orthus' average performance.

repeated 10 times, and report their minimum and maximum computation times for both tasks in Fig. 11. Please note MetaOD was not able to be tested with this methodology as a single instance of its fitting consumed more than two hours of computation time. In our anecdotal tests, MetaOD's recommendation phase computation is inferior to 100 ms.

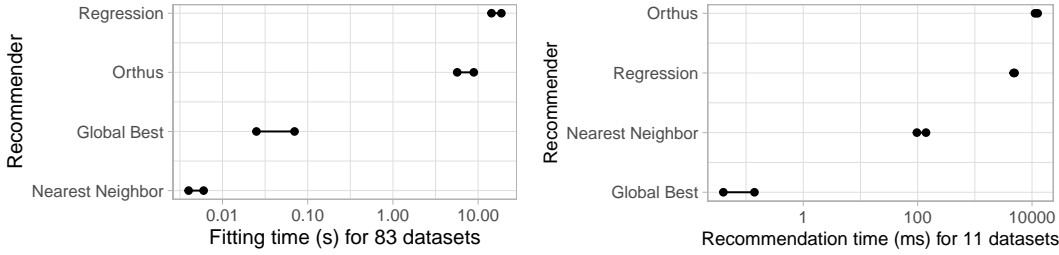

Figure 11: Fitting and recommendation times for the tested recommenders.

