# OpenReview forum: "Meta-Learning for Fast Model Recommendation in Unsupervised Multivariate Time Series Anomaly Detection "
_automl.cc/AutoML/2023/Conference — AutoML 2023 MainTrack_

### Official Review · Reviewer_dJd2 · 2023-03-29

**Potential Impact On The Field Of Automl Rating:** 3
**Technical Quality And Correctness Rating:** 4
**Clarity Rating:** 3
**Actions Required To Increase Overall Recommendation:** Look at the weakness section.

**Summary Of Contributions:**

The authors provide a "recommender system" called hydra to predict which anomaly detector to use on a new dataset, given the past performance data of the set of detectors on other datasets.
Their main contributions lie in
a. applying an anomaly-specific set of meta-features
b. identifying clusters in the (projected) meta-feature space and selecting a recommender that best suits this cluster.
c. identifying if a new dataset is not in the vicinity of the training distribution. In that case



**Clarity:**

I am not entirely sure about this one, but I recall, that not all projections need to be distance-preserving. Given that the DBSCAN clustering and the subsequent cluster assignment hinge on the distance in the UMAP projection, I would appreciate it if the authors could clarify this aspect and potential implications for their methodology.

3.3.1 How do you configure DBSCAN - as the number of clusters and their circumferences are dependent on its hyper parameterization?
How do you deal with very small clusters and the little data used to train the respective recommender (3.3.3 L131-133)? Please report the found cluster sizes.

3.3.2. Motivate, how in case of radically differing meta-features from the training distribution, we can at all make a sensible prediction (given that they by definition have little variability we can confidently exploit for prediction)

3.3.3 Please reformulate your assumption in
"Given the inherent variety in the
potential input datasets we could find, it’s reasonable to assume that existing patterns between
 the meta-features and performance matrices can vary across the meta-feature space. Otherwise
 stated, we expect that datasets that fall in close region of the meta-feature can provide a better
input for recommendation than faraway datasets." for better clarity.

L133-136: when you say you are having n tuples, you refer to the number of configurations right?

Please elaborate on the data sources. If I understand Figure 1 and the description in 4.4 correctly, you are stratifying across benchmarks.



**Overall Review:**

Strengths:
1. The approach can distinguish in the meta-feature space whether it has encountered similar datasets (based on the distance in the projected meta-feature space). This way, the model can exploit more specialized recommenders for particular sets of the meta-feature space.

Weaknesses
1. I would like to start by pointing out that the name of the method Hydra is not well motivated and that it easily is confused with a widely used python package [hydra.cc](hydra.cc) which is used for configurable experimentation. I recommend changing the name.

2. The definition in L.44 of what meta-learning is appears to be wrong. Instead, this seems to be a description of algorithm selection.
This is also one of my concerns regarding how the paper is situated; while technically correct, that a recommendation is made and similarities exist in this line of thought, the entire outline rather strikes me as an algorithm selection problem. This phrasing is more easily referable in this research area. Crucial here is that the configuration (algorithm and its hyperparameters) is a discrete set for which we want to predict a performance given some meta-experience (performance data) with that set. Also, the section headers and paragraphs 3., 3.1, 3.2 simply describe an algorithm selector. This description also facilitates the unfortunately complex title "Unsupervised Model Recommendation through Meta-Learning"

3. As with most algorithm selectors, the dependency on meta-features eventually limits the performance of a selector by the quality of these features and how well they are "aligned"/indicative of the performance. The time-series aspect of this paper can mostly be reduced to a selection of time-series targeted meta-features. In effect, this opens up a lot of baselines from the AS literature to compare against.

4. I acknowledge, that in AS, datasets are hard to come by, but I wouldn't go as far as claiming that 4 benchmarks are an unseen extent.

5. A potentially important hyperparameter regarding the complexity in both functional expressivity and cost of this method lies in the number of singular vectors. Another set of important hyperparameters are those regarding a) the UMAP projection and b) the decision on how to cluster the meta-feature. Effectively, this will change the model's performance.

**Potential Impact On The Field Of Automl:**

From my point of view, it is an algorithm selector (amongst many in the existing literature) so this community likely will cite the paper eventually. Knowing when a recommendation is good and reliable is important to give the user confidence. This paper provides a means of finding regions in the meta-feature space that are more populated and one would expect that the recommendation is more likely to be accurate. Research in the confidence of the recommendation is something that would be valuable.

**Review Confidence:**

4: You are confident in your assessment, but not absolutely certain. It is unlikely, but not impossible, that you did not understand some parts of the submission or that you are unfamiliar with some pieces of related work.

**Review Rating:**

4: Weak Reject: For instance, a paper with minor technical flaws, limited impact, and/or weak evaluation.

**Review Summary:**

I would think that this paper should be placed in the algorithm selection setting and may require considerable rewriting (that using this perspective will simplify the paper) and also require more baselines from the AS field. Any potential empirical gain should be demonstrated against those baselines.

**Technical Quality And Correctness:**

I am relatively confident in the technical correctness, albeit AS methods are likely baseline candidates.

---

> ### Author Response · Authors · 2023-04-29
> **Answer to reviewer's comments (part 1)**
>
> We are deeply thankful to the reviewer for their comments.  We have uploaded a new version of the article with the changes highlighted in blue to facilitate the review task. We proceed to answer several of the comments below:
>
> **Clarification on DBSCAN when using UMAP**
>
> The reviewer is right in that UMAP doesn't preserve distances and this can be an issue when performing clustering in its space. We believe in our case it is not harming the model (and we include explanations in sections 3.3.1 and 3.3.3): for Hydra's novelty detection step, the only decision to make is whether the new point is close to any existing point or not. Since neighbor distances are relatively well preserved, this is still possible. For CFact's internal clustering, it may produce smaller clusters than needed ([https://umap-learn.readthedocs.io/en/latest/clustering.html](https://umap-learn.readthedocs.io/en/latest/clustering.html)), but it will not  join points that are not close in the original space, so the intended effect of CFact (separating configurations with different performance profiles in order to improve URegression's performance) will still be there.
>
> **How do you configure DBSCAN - as the number of clusters and their circumferences are dependent on its hyper parameterization?**
>
> It is definitely true that the clustering step will impact the recommender's performance. In our tests, Hydra was quite robust against different values of DBSCAN; nevertheless, we will include an ablation study for different clustering hyper parameters in the appendix to make this explicit.
>
> **How do you deal with very small clusters and the little data used to train the respective recommender (3.3.3 L131-133)? Please report the found cluster sizes.**
>
> We do apologize but we cannot report the found cluster sizes in a comprehensive manner; due to the cross-validation process, 150 clustering processes have been run just for Scenario I, which is not possible to report fully, and pointless to report cursorily (as a compromise, we will report a statistical summary in the appendix). Regarding the problem of small clusters, they are not necessarily an issue: a small cluster in CFact means a partition of a small number of configurations of the performance matrix, but every dataset was evaluated on each of them, so we have full performance profiles for each algorithm inside the cluster.
>
> There are still two potential pitfalls we see regarding small clusters:
>
> 1.  the noise cluster produced by DBSCAN will not be associated to meaningful patterns (i.e., which correspond to configurations that behave dissimilar to others and from which URegression would not be able to generalize); at the same time, as per our overall results, the incidence of noise cluster does not seem to affect results in a significantly negative way.
> 2.  Small cluster could have a small technical impact for SVD computation, as SVD cannot be performed for the requested number of singular vectors: however, as URegression employes 2 (or 5) singular vectors, only clusters smaller than 2 (or 5) could constitute a problem, which could be easily circumvented by setting minPoints in DBSCAN to 2 (or 5).
>
> Incidentally, we use default DBSCAN hyperparameter, for which minPoints equals 5 and as such we do not experience this technical problem.
>
> **3.3.2. Motivate, how in case of radically differing meta-features from the training distribution, we can at all make a sensible prediction (given that they by definition have little variability we can confidently exploit for prediction)**
>
> We have included a sentence at the end of 3.3.2 detailing that a low number of singular vectors extracted from SVD forces the method to focus on only the broadest, most general patterns, which we expect to be applicable across novel data too.
>
> **L133-136: when you say you are having n tuples, you refer to the number of configurations right?**
>
> This was a typo, it should have been the number of clusters (instead of the number of datasets), as the number of produced tuples at the output of CFact is equal to the number of clusters found / URegression recommenders trained. Each output is of the form {recommended configuration, estimated performance}. Thanks for spotting the nit, we have fixed it.
>
> **On Hydra's name**
>
> We thank the reviewer for the pointer, as we were not aware of the existence of the aforementioned package. If the reviewer thinks it's convenient, we could change the recommender's name, in the same vein, to another multi-headed monster of mythology: Orthos. We have not changed the figures and text for this rebuttal period due to time constrains, but we are totally willing to do so for the final version.

---

> > ### Author Response · Authors · 2023-04-29
> > **Answer to reviewer's comments (part 2)**
> >
> > **Regarding meta-learning, algorithm selection and other potential methods**
> >
> > The reviewer is right in the sense that this paper could be cast in the broader area of Algorithm Selection, that we refer to as Model Recommendation. In this area, we decided to align with what the literature refers to as meta-learning (see e.g., Joaquin Vanschoren. Meta-Learning: A Survey. [http://arxiv.org/abs/1810.03548](http://arxiv.org/abs/1810.03548), Yue Zhao et al. Automatic Unsupervised Outlier Model Selection, [https://proceedings.neurips.cc/paper/2021/file/23c894276a2c5a16470e6a31f4618d73-Paper.pdf](https://proceedings.neurips.cc/paper/2021/file/23c894276a2c5a16470e6a31f4618d73-Paper.pdf)), to differentiate from other methods that do not require previous labelled data but do require the application of every possible configuration considered in the new dataset (see [https://www.andrew.cmu.edu/user/yuezhao2/papers/23-kdd-explorations-ipm.pdf](https://www.andrew.cmu.edu/user/yuezhao2/papers/23-kdd-explorations-ipm.pdf)). The meta-learning settings are more fit to our scenario, as fitting >500 anomaly detectors is not feasible every time a new dataset is received.
> >
> > The semantic difference between algorithm selection and meta-learning is even acknowledged in the wikipedia page for Algorithm Selection (AS), written by the Coseal group: [https://en.wikipedia.org/wiki/Algorithm_selection](https://en.wikipedia.org/wiki/Algorithm_selection)). The reviewer's remark is completely right, though: while we compared against the state of the art in the meta-learning part, other Algorithm Selection methods could be compared against ours. We have included so in the Conclusions and it opens a broader research and comparison area. We thank the reviewer for this remark.

---

> > > ### Comment · Reviewer_dJd2 · 2023-05-02
> > > **Final Comment**
> > >
> > > Thank you for your efforts in revising the paper.
> > >
> > > Thanks for reporting the DBSCAN HP ablation. I found this helpful and also supports my intuition, That DBSCAN's HP can indeed affect your method, albeit it is mostly the dynamic range in the performance that changes along the Hyperparameters. More seeds than one would make this a more reliable ablation, though.
> > >
> > > A few pointers,
> > >
> > > I am still not entirely convinced of the UMAP-smaller-cluster argument. As you and its maintainers point out, it is cumbersome to use and requires considerable validation on the cluster level. As much as I appreciate UMAP for its elegance, I myself have been stumbling over this at some point and instead used UMAP as a visual validation process of another clustering method instead, as it was less prone to concerns that I now myself am issuing.
> > > Considering, that smaller clusters are more likely to produce unstable recommendations based on the little data that is available for this subspace - which would be "smoothed over" for larger clusters, this surely can affect the recommender's performance and should be dealt with carefully.
> > >
> > > I find the Orthos name a better choice. But maybe there is something more tangible, more related to the method or its components.
> > >
> > > Overall, I would lean towards weak acceptance.

---

### Review · Reproducibility_Reviewer_NtaY · 2023-04-02

**Completeness Of Code And Dataset Supplement Rating:** 3
**Usability And Ease Of Reproducibility Rating:** 2
**Actions Required To Increase The Reproducibility And Overall Recommendation:** Mention the R and Rstudio version.

**Completeness Of Code And Dataset Supplement:**

They include code, but they mention that they don't include all data. I would like to understand why they do not include all data.

**Overall Reproducibility Review:**

I cannot reliably assess the reproducibility cause I ran into errors configuring the environment. The errors are not caused by the author's work.

**Review Confidence:**

1: You are unable to assess this paper’s reproducibility and have alerted the ACs to seek an opinion from different reviewers.

**Review Rating:**

6: Borderline: Leaning Accept, all critical aspects are reproducibile with minor effort, and the remainder are likely reproducible with major additional effort.

**Review Summary:**

The code and the reproducibility are difficult to assess for this reviewer.

**Summary Of Necessary Code And Dataset Supplement:**

The code is in R, which I am not familiar to, thus I cannot judge how clean it is. The authors include nice instructions about how to reproduce the results. I tried to run it but I had issues configuring R-studio. I would value it if the authors provide more information about the R and R-studio version.

**Usability And Ease Of Reproducibility:**

It is not easy to reproduce. The code is in R programming language which makes the environment creation unsual for this reviewer. Moreover, some important information for reproducbility is missing  ( R version).

---

> ### Author Response · Authors · 2023-04-29
> **Answer to reproducibility reviewer's comments**
>
> We thank the reviewer for their efforts. We reviewed the code and there was, indeed, a bug in the instructions. We've revamped the code to simplify the components to install, the interaction needed and included clearer instructions to launch the code. Now the environment is managed automatically and the reviewer only has to install R's latest version (and Rtools if tested on Windows).

---

### Official Review · Reviewer_fynp · 2023-04-05

**Potential Impact On The Field Of Automl Rating:** 3
**Technical Quality And Correctness Rating:** 3
**Clarity Rating:** 4

**Summary Of Contributions:**

The authors presenta meta-learning based approach for AutoML on unsupervised anomaly detection in multivariate time series. The system `Hydra` is performs favorably copmpared to SoTA approaches, especially for novel or unseen data sources.
The architecture consists of (i) novelty detector (ii) model recommenders based on novelty in the meta-feature space. The recommender for novel  data relies on SVD and Random Forest to rank models.
A large number of datasets are utilized for the meta-training phase, and catch22 is used for computing meta-features. A wide set of anomaly detection approaches are covered.Performance is demonstrated to be better than the current SoTA. The authors present a fairly comprehensive evaluation and ablation study as well.

**Actions Required To Increase Overall Recommendation:**

Provide computational overhead figures in a more comprehensive manner.


**Clarity:**

The clarity is overall good, though a few areas could be improved. Pseudocode or Algorithm figure would have helped  in clearly communicating ideas to the reader.  The writing fairly clean and crisp, and the narrative is fluid. There are a few figures, and that could be improved but the space constraint is understandable.



**Overall Review:**

The paper has many strong points.
1. The core idea is robust, based on meta learning. The whole architecture with the novelty detection (of data) component and the recommender heads are well founded and make intuitive sense.
2. A wide ranger of anomaly detection algorithms are covered in the meta learning process. Also, many datasets are used, and hyperparameter tuning is done.
3. Empirical evaluation and analysis is performed adequately. The ablation study (Sec 5.3.1, Sec 5.3.2) is useful in providing the user understanding.

Negative aspects:
1. Some statements are not very clear. For instance Sec 5.3.1,Line 274-Line 276.
2. Sec 5.4 provides computational requirements in an anecdotal fashion. A clear figure or performance numbers would be effective in this regard.
3. Some design choices could be better explained. e.g. use of catch22.
4. How are hyperparameters tuned for the recommended algorithm is not clear.
5. Meta learning based approaches (such as AutoSklearn 1, 2) have been successful. The authors concede that their system requires more data for better performance. How does that limit the utility of the current iteration is not clear. The concern that there is inadequate data is not addressed.

**Potential Impact On The Field Of Automl:**

The paper would be widely useful, as it addresses a practical and prevalent problem.  So the impact would be positive, and it would be cited.

**Review Confidence:**

3: You are fairly confident in your assessment. It is possible that you did not understand some parts of the submission or that you are unfamiliar with some pieces of related work.

**Review Rating:**

7: Weak Accept: Technically sound paper with moderate-to-high impact and strong evaluation, with perhaps some minor flaws.

**Review Summary:**

The paper presents an effective solution for a interesting and practical problem. The paper presents some good insights , and is well written and the formulation is sound. However a few points that need to be addressed remain as noted.

**Technical Quality And Correctness:**

The technical quality of the paper is good.  The idea of the paper follows from meta-learning based approach to AutoMl, which has seen some success. Yet the performance of these systems are reliant on the meta datasets. Also anomalies, by definition are scarce and application specific. So the quality of the meta datasets are crucial.

The design architecture is sensisble and well explained. The approach on data novelty using meta-features and dimensionality reduction makes sense. But meta-features for time series are somewhat less well understood, and while the intuition makes sense the actual efficacy may be lower. Time series similarity computation has been shown to have issues (Refer: https://doi.org/10.1007/s10115-004-0172-7)

One point that the authors mention is that catch22 is efficient is not well supported by the authors (maybe cite something like : doi:10.1109/ICDMW53433.2021.00134). Since (anecdotally) in real world datasets it takes quite a bit of time (compared to other libraries such as tsfresh etc). Why is catch22 chosen vs. hctsa, Kats or TSFEL needs more explanation. Is it due to presence of 22 important meta features ? Also catch22 is for univariate time series, how is it adopted to multivariate case is not well explained (apart from 110 features are taken).

---

> ### Author Response · Authors · 2023-04-29
> **Answer to reviewer's comments**
>
> We are very thankful for the reviewer's comments, we proceed to answer to some of them below. We have uploaded a new version of the article with the changes highlighted in blue to facilitate the review task.
>
> **One point that the authors mention is that catch22 is efficient is not well supported by the authors (maybe cite something like : doi:10.1109/ICDMW53433.2021.00134). Why is catch22 chosen vs. hctsa, Kats or TSFEL needs more explanation. Is it due to presence of 22 important meta features? Also catch22 is for univariate time series, how is it adopted to multivariate case is not well explained (apart from 110 features are taken).**
>
> We thank the reviewer for the reference and we have included it, as well as three reasons to use Catch22 (their performance in ML tasks, their feature independence and their computation time) in section 3.4. We have also added a clarification on how the 110 meta-features are extracted, in the same section. Finally, we added a discussion on future work concerning on meta-features in Section 6.
>
> **Pseudocode or Algorithm figure would have helped in clearly communicating ideas to the reader.**
>
> We agree with the reviewer that details about several important aspects of Hydra design were omitted in the previous explanation. To improve the description, we have included diagrams for both base methods, URegression and CFact (Figures 1 and 2) and reworded the intuition and explanation of CFact. We hope this will make understanding them easier.
>
> **Some statements are not very clear. For instance Sec 5.3.1,Line 274-Line 276.**
>
> We agree and have reformulated it to be clearer.
>
> **How are hyperparameters tuned for the recommended algorithm is not clear.**
>
> The recommender's output is a configuration itself, being an algorithm + specific hyperparameter values, so no further tuning is needed after recommendation. When choosing the different values to generate our configuration library, we tried to use large enough ranges to cover potential different behaviors in the algorithms.
>
> **Meta learning based approaches (such as AutoSklearn 1, 2) have been successful. The authors concede that their system requires more data for better performance. How does that limit the utility of the current iteration is not clear. The concern that there is inadequate data is not addressed.**
>
> The reviewer is right in that we had not touched on this. We highlighted it in the discussion, including two references: one highlighting problems with benchmarks and another one referring to improved metrics for time series anomaly detection.

---

> > ### Comment · Reviewer_fynp · 2023-05-05
> > **Addressing of Concerns**
> >
> > 1. [+] The authors have adequately addressed the concern regarding catch-22.
> > 2. [-] However, clarity is still lacking. Diagrams  only do so much, a clear algorithmic structure is needed to concisely convey the idea. Also, descriptions are still somewhat vague. For instance, in Figure 2 the caption ".. 𝑢𝑚𝑎𝑝 a dimensionality reduction of the transposed performance matrix..."--- UMAP is a widely known method, and as for notations using a single letter is more standard practice. The rewording and additional clarity helps, but is not sufficient.
> >
> > Also Hydra is eponymous with a very popular configuration handler, and should not be used.

---

### Official Review · Reviewer_6xX9 · 2023-04-11

**Potential Impact On The Field Of Automl Rating:** 3
**Technical Quality And Correctness Rating:** 3
**Clarity Rating:** 4

**Summary Of Contributions:**

This paper proposes Hydra, a meta-learning-based model recommender for anomaly detection in multivariate time series data. The authors highlight the real-world restrictions that make existing methods unusable, and address the need for generalization to previously known or unseen data sources. The proposed approach outperforms the current state of the art, achieving a higher performance even in difficult scenarios where data similarity is minimal. The experiments are conducted on a large number of public datasets from different data sources, and the ablation study confirms the efficacy of Hydra in recommending models in the top 10% for known sources of data and the top 13% for unseen sources. This work is an important contribution to the field of unsupervised model recommendation for anomaly detection and provides a promising direction for future research in this area.

**Actions Required To Increase Overall Recommendation:**

since the work claim that their work can be applied for real-world time series data, they should evaluate Hydra in a larger dataset and more complicated data distributions

**Clarity:**

this paper is well written and easy to follow. discuss more on the above-mentioned third type of data would be better.

**Overall Review:**

Positive aspects:

The paper proposes Hydra, the first meta-recommender for anomaly detection in multivariate time series data, which is a novel contribution to the field of unsupervised model recommendation.
The paper addresses real-world restrictions, such as limited time to issue a recommendation and the need for generalization to previously unseen data sources, which are important considerations in practical applications.
The paper conducts a thorough experimental evaluation using a large number of public datasets from different data sources, and provides extensive analysis of the results.
The paper clearly presents the proposed approach, including the use of meta-learning and the design of Hydra, making it easy for readers to understand the methodology.

Negative aspects:

The paper could benefit from providing a more detailed discussion of the limitations and challenges of existing methods for unsupervised model recommendation for anomaly detection in multivariate time series data, to help readers better understand the motivation for the proposed approach.
The paper could provide more information on the specific design of Hydra, such as how the meta-learning framework is applied to the problem of model recommendation for anomaly detection, and how the proposed approach addresses the real-world restrictions and need for generalization to previously unseen data sources.
The paper could benefit from providing details of dataset split (train/val/test)

**Potential Impact On The Field Of Automl:**

The paper's contributions are likely to be important for the field of AutoML, particularly in the area of unsupervised model recommendation for anomaly detection in multivariate time series data. The proposed approach using meta-learning and Hydra, the first meta-recommender for anomaly detection in literature, provides a novel and effective solution to the problem of model selection. The experimental evaluation on a large number of public datasets from different data sources adds to the credibility of the proposed approach. The paper's contributions are likely to be of interest to researchers working in the field of AutoML, specifically in the area of unsupervised model recommendation. As such, the paper is likely to be cited by others in the field who are interested in this topic.

**Review Confidence:**

3: You are fairly confident in your assessment. It is possible that you did not understand some parts of the submission or that you are unfamiliar with some pieces of related work.

**Review Rating:**

7: Weak Accept: Technically sound paper with moderate-to-high impact and strong evaluation, with perhaps some minor flaws.

**Review Summary:**

Based on my assessment, I recommend accepting the paper with minor revisions. The paper proposes Hydra, a meta-recommender for anomaly detection in multivariate time series data, and conducts a thorough experimental evaluation to demonstrate its effectiveness. The proposed approach addresses real-world restrictions and provides a novel contribution to the field of unsupervised model recommendation. While there are some areas where the clarity could be improved, including a more detailed discussion of existing limitations and challenges, providing more information on the specific design of Hydra, and more detailed analysis of the experimental results, these issues can be addressed with minor revisions. Overall, the paper provides a valuable contribution to the field and warrants acceptance with minor revisions.

**Technical Quality And Correctness:**

The proposed method is intuitive. It proposes two heads, one for known data, the other fro unseen data. However, there is still a third case in which the data is composed of a few known data and a lot of unseen data. This work does not seem to solve this case.

---

> ### Author Response · Authors · 2023-04-29
> **Answer to reviewer's comments**
>
> We express our gratitude to the reviewer, their comments have allowed us to clarify and improve our work. We have uploaded a new version of the article with the changes highlighted in blue to facilitate the review task. We proceed to answer to some of the comments issued:
>
> **There is still a third case in which the data is composed of a few known data and a lot of unseen data. This work does not seem to solve this case.**
>
> We may not have been clear in our explanations, but the recommendation is performed per dataset. So when queries for new datasets are received, in which some of the datasets are close to known datasets (i.e., datasets known at training time) and others are novel (akin to unseen data), each of the query would be resolved by the best corresponding recommender independently (i.e., Hydra would select CFact for the former and UReg(2) for queries of the latter kind). Our recommender does not inspect internal patterns of the data to characterize those as known or unknown, only the distance of the whole dataset's meta-features to existing ones.
>
> Of course, when queries are mostly for unseen data, then the value of the training is reduced (i.e., whether CFact had previous experience on similar data), but nevertheless Hydra in this case relies on UReg(2) which as proven by our leave-one-provider-out (Scenario II) experiments improve with respect to the state of the art. As part of our future work, we plan to further extend the experimental campaign to even more datasets, which could allow us to explore more deeply the interesting issue raised by the reviewer -- which is however a knowingly difficult task owing to the lack of public available multi-variate time-series data with ground-truth. We added a comment in Section 6 concerning this aspect.
>
> **The paper could benefit from providing a more detailed discussion of the limitations and challenges of existing methods for unsupervised model recommendation for anomaly detection in multivariate time series data, to help readers better understand the motivation for the proposed approach.**
>
> We agree we were not clear enough. We have added a summary at the end of section 2 clarifying the main differences between our work and previous proposals, namely, the complete lack of focus on time series in either the methodological design (including meta-feature selection) or algorithmic space tested, as well as a lack of attention to novelty detection, which we consider crucial, as existing methods may overfit on local patterns that may not generalize well.
>
> **The paper could provide more information on the specific design of Hydra, such as how the meta-learning framework is applied to the problem of model recommendation for anomaly detection, and how the proposed approach addresses the real-world restrictions and need for generalization to previously unseen data sources.**
>
> We agree with the reviewer that details about several important aspects of Hydra design were omitted in the previous explanation. To improve the description, we have included diagrams for both base methods, URegression and CFact (Figures 1 and 2) and reworded the intuition and explanation of CFact.
>
> Additionally, to clarify that we are measuring generalization capabilities, we have expanded the end of Section 4, highlighting that the Scenario II addresses real-world restrictions, since the Hydra recommender has been trained on totally different data, and no single dataset in the training data comes from the same distribution or provider as the test data.
>
> **The paper could benefit from providing details of dataset split (train/val/test)**
>
> While ultimately, for repeatability purposes,we believe that the code we provide is the ultimate source for these details, we have added some clarification in Section 4.4 for Scenario I (10-fold cross validation with 90%/10% train/test split) and Scenario II (4-fold leave-one-provider-out validation, with different train/test splits in this case since the size of folds in this case depends on the heterogeneous size of each provider data).
>
> **Since the work claim that their work can be applied for real-world time series data, they should evaluate Hydra in a larger dataset and more complicated data distributions.**
>
> We definitely agree with the reviewer and this is indeed part of our plans: further extending the experimental campaign to even more datasets, would allow us to explore more deeply the generalization ability of Hydra. At the same time, we hope the review concur with us that this is knowingly difficult task owing to the lack of public available multi-variate time-series data with ground-truth: as far as we know, we already use every publicly available multi-variate time-series data including labeled anomalies. This is a known problem, which requires a community-wide effort.  We added a comment in Section 6 concerning this aspect.

---

### Official Review · Reviewer_attc · 2023-04-12

**Potential Impact On The Field Of Automl Rating:** 3
**Technical Quality And Correctness Rating:** 3
**Clarity Rating:** 4
**Actions Required To Increase Overall Recommendation:** ...

**Summary Of Contributions:**

The authors propose Hydra, a meta-learning recommendation framework for novelty detection. Hydra uses two inner recommender systems to deal with: known and novelty data. The tool has an inner mechanism that automatically actives which recommender should be used.

**Clarity:**

* Which features are in the C22 set? How are they applied to multivariate datasets?

* Figure 2 is not a figure. It is a table. Change it, please.

* Also, captions could be provided above the Figure for readability.

* Suggestion: Add a figure/diagram with the Hydra flow (online/offline tasks) and other components in the Methods section's beginning. It would help to understand its flows better.

**Overall Review:**

The paper proposes a new meta-recommender and presents a nice comparison to some related works in the specific area (meta-learning recommenders for novelty detection and time series). The methodology is also well described, and results are organized according to the research questions and present statistical evaluations. Results also present good implications for discussing Hydra's performance against simpler baselines and MetaOD.

On the other hand, it is missing comparisons versus cited baselines (LOTUS, ELECT). The authors of these studies could be contacted by email and asked about their tools.

**Potential Impact On The Field Of Automl:**

The paper has a good impact since it presents a new approach to model selection for anomaly detection. Of course, considering all of its implications, it needs further investigations, but it is valuable work.

**Review Confidence:**

4: You are confident in your assessment, but not absolutely certain. It is unlikely, but not impossible, that you did not understand some parts of the submission or that you are unfamiliar with some pieces of related work.

**Review Rating:**

8: Accept: Technically sound paper with major impact and strong evaluation, with perhaps some minor flaws.

**Review Summary:**

Based on the previous comments, I think the paper presents a good study, still in its initial, but with many questions that can contribute to the AutoML area in time series. Thus, my recommendation is to accept the paper.

**Technical Quality And Correctness:**

The paper is well-structured and mostly understandable. The single question is regarding results when comparing Hydra x MetaODS. Figure 3 shows a considerable difference in performance between them in Scenario II. Why does this occur? Is it conditioned to the experimental scenario (datasets, folds) or something else?

---

> ### Author Response · Authors · 2023-04-29
> **Answer to reviewer's comments**
>
> We thank the reviewer for their comments and insights. We have uploaded a new version of the article with the changes highlighted in blue to facilitate the review task.
>
> **The single question is regarding results when comparing Hydra x MetaODS. Figure 3 shows a considerable difference in performance between them in Scenario II. Why does this occur? Is it conditioned to the experimental scenario (datasets, folds) or something else?**
>
> Our hypothesis is that Hydra is able to get a better result thanks to the dynamic selection of recommenders: CFact creates hyper-specialized recommenders, but URegression with 2 factors is forced to keep only the broadest and most relevant patterns across datasets, so they're able to be exported across data sources. MetaOD is similar to URegression but done with a larger number of singular vectors, which forces it to focus on finer details and patterns that don't translate across data sources. As such, MetaOD and CFact performance are very close in Scenario I, while UReg(2) is much better than MetaOD at generalization is Scenario II: overall,  meta-recommendation by Hydra is able to select the best among CFact and UReg across all scenarios.
>
> **Which features are in the C22 set? How are they applied to multivariate datasets?**
>
> Regarding Catch22, while we cannot include a full detail of the contained features in the main paper due to space constrains, we include it in appendix E. We have also expanded Section 3.4 to clarify how they're used in a multivariate setting, and expanded considerations about alternative or extensions to Catch22 and related summarization in Section 6.
>
> **Figure 2 is not a figure. It is a table. Change it, please.**
>
> We have corrected this and that element is now a table.
>
> **Suggestion: Add a figure/diagram with the Hydra flow (online/offline tasks) and other components in the Methods section's beginning. It would help to understand its flows better.**
>
> We added diagrams for both URegression and CFact (Figures 1 and 2), as these are the basic building blocks of Hydra, and the core of our contribution. We have also reworded CFact's explanation to ensure the recommenders design is clear.

---

### Official Review · Reviewer_aCMM · 2023-04-18

**Potential Impact On The Field Of Automl Rating:** 3
**Technical Quality And Correctness Rating:** 2
**Clarity Rating:** 2

**Summary Of Contributions:**

The authors propose a 2 level meta-learning approach for recommendation of algorithm configurations in multivariate time series anomaly detection. At the first level, the decision is if a new time series is similar to the ones used for training. Depending on the result, a different meta-model is used to make the recommendation.

The evaluation separates two scenarios: new time series from known domain and new domain. Both scenarios make sense but the second is not usually considered.

The results show that the approach competes with the state-of-the-art and that the two-level approach is better than the two base-level recommendation models.

**Actions Required To Increase Overall Recommendation:**

Explain and motivate the method more clearly.


**Clarity:**

The motivation for having a 2-level approach is clear. However, the motivation for the use of distances to choose the base-level method should be further investigated. Furthermore, the reason why the methods used at the base-level were chosen is also not clear.

The description of the motivation for the ablation study is not clear. However, the reasoning underlying the discussion of the results is.

Furthermore, it is not clear:
-how the PRAUC values are transformed
-why standard statistical comparison tests were not used for significance of differences


**Overall Review:**

The idea of separately handling similar and different time series in multivariate scenarios, both in modelling and evaluation, is quite interesting and should be followed.

However, even though the method is interesting, the motivation for the solutions incorporated into it are not entirely clear to me.

On the other hand, the evaluation methodology is quite interesting and should be followed in empirical studies concerning recommendation of algorithms multivariate time series.

More detailed comments are provided above.

Other comments:
-couldn't UReg optimize for ranking rather than for error?
-fig 2 not references in text


**Potential Impact On The Field Of Automl:**

The idea of separately handling similar and different time series in multivariate scenarios, both in modelling and evaluation, is quite interesting and should be followed.


**Review Confidence:**

4: You are confident in your assessment, but not absolutely certain. It is unlikely, but not impossible, that you did not understand some parts of the submission or that you are unfamiliar with some pieces of related work.

**Review Rating:**

7: Weak Accept: Technically sound paper with moderate-to-high impact and strong evaluation, with perhaps some minor flaws.

**Review Summary:**

Although I'm not entirely convinced by the method, I find the general approach interesting, particularly in what concerns evaluation.


**Technical Quality And Correctness:**

The 2 evaluation scenarios make evaluation of predictive methods in multivariate time series more realistic. However, the use of benchmark datasets for anomaly detection in time series as recently been questioned (https://arxiv.org/abs/2009.13807). This should be discussed by the authors.

Concerning the method proposed there are important technical issues:

1. Although the method is presented as unsupervised, the offline training phase is supervised.

2. Although the method is presented as suitable for multivariate time series, it essentially depends on the distance between time series, which is not necessarily related to the domain (i.e. similar time series are not necessarily from the same domain; dissimilar ones are not necessarily from different domains). The relation between distance and domain is never really analyzed.

3. Many data characterisation approaches use label information, which, in the current approach, is available in the offline training but not at online application. This issue should be discussed and it should be clear that the metafeatures cannot use labels.

4. The multivariate metafeatures only describe very basic characteristics of the distribution of basic metafeatures across the different time series of a dataset. Given that the problem is anomaly detection, I would expect metafeatures that characterise deviations from the distribution.

---

> ### Author Response · Authors · 2023-04-29
> **Answer to comments by reviewer (part 1)**
>
> We thank the reviewer for their insightful comments. We proceed to answer to it below, per comment:
>
> **The use of benchmark datasets for anomaly detection in time series as recently been questioned. This should be discussed by the authors.**
>
> We agree that the quality of benchmarks is, indeed, a recently discussed problem. While the available data for our scenario (multivariate industrial time series) is limited, we tried to ensure the potential effects of bad labelling was limited, by only including one of the discussed benchmarks (SMD), which represents less than 30% of our datasets. Nevertheless, we've included its potential effect as a limitation in the discussion, as well as another reference that cites the paper the reviewer suggested regarding potential issues with the metric we used, which opens a future work exploring the effect of different metrics on our task. We thank the reviewer for the pointer.
>
> **1. Although the method is presented as unsupervised, the offline training phase is supervised.**
>
> Indeed, this is a limitation of the meta-learning approaches that we had to adopt in order to ensure our recommendations can be done in the order of seconds. Nevertheless, the recommendation itself does not require labels on new data, which makes it unsupervised.
>
> **2. Although the method is presented as suitable for multivariate time series, it essentially depends on the distance between time series, which is not necessarily related to the domain. The relation between distance and domain is never really analyzed.**
>
> We'd like to clarify the method depends on distance in the meta-feature space, not in the original time series space, which makes it more flexible than just comparing in the original space. Regarding the relationship between domain and distance in the meta-feature space, we display it on Figure 3, where we show that data from the same sources do tend to be clustered together.
>
> **3. Many data characterisation approaches use label information, which, in the current approach, is available in the offline training but not at online application. This issue should be discussed and it should be clear that the metafeatures cannot use labels.**
>
> We agree it's a needed clarification. We have added "for which no labels are needed" in 3.1 when describing the concept of meta-features to emphasize their unsupervised nature.
>
> **4. The multivariate metafeatures only describe very basic characteristics of the distribution of basic metafeatures across the different time series of a dataset. Given that the problem is anomaly detection, I would expect metafeatures that characterise deviations from the distribution.**
>
> We agree other summarization features could be extracted from the meta-feature set.  The current ones include measures of centrality (average, Q1 and Q3) and outliers (minimum and maximum). As the regression method we use is a Random Forest, interactions between these two families should be captured by the decision trees, which would represent the behavior the reviewer is expressing. Nevertheless, this is still an open area on which we only used one possible option and so we acknowledge it in section 6 when regarding future work about meta-features.
>
> **Motivation for the use of distances to choose the base-level method should be further investigated / The reason why the methods used at the base-level were chosen is also not clear.**
>
> We agree that some of the explanations and reasonings about the methods weren't all clear: we've revamped their sections, including diagrams for both URegression and CFact (Figures 1 and 2), as well as improved the justification and definition of CFact.
>
> **It is not clear how the PRAUC values are transformed**
>
> We agree, we have included an example of the transformation and reworded the explanation to make it clearer.
>
> **It is not clear why standard statistical comparison tests were not used for significance of differences**
>
> This is something that we discussed internally extensively. We opted to use meta-analysis techniques for three reasons: 1. We had to perform thousands of pairwise comparisons between recommenders (20 per fold), which would increase our Type I error and force us to use an extremely strict Bonferroni correction of the p-values, 2. p-values don't reflect a measure of the effect size, in this case the difference in performance between algorithms, which we considered key to display and 3. For space and clarity, summarizing hundreds of p-values across several dimensions was not something we could do given the available page count. Nevertheless, we add a table of these p-values in appendix G for completeness.

---

> > ### Author Response · Authors · 2023-04-29
> > **Answer to comments by reviewer (part 2)**
> >
> >
> > **Couldn't UReg optimize for ranking rather than for error?**
> >
> > Totally true, and it is in fact what MetaOD (the state of the art method we compare against) does. We didn't do it for two reasons: 1. in our tests, URegression's performance was lower when using it and 2. if it optimizes per rank it's impossible to use it in CFact, as the scores from independent UReg recommenders wouldn't be comparable.
> >
> > We hope the changes we've detailed and our explanations in this comment satisfy the reviewer's petition and thank them again for their deep insights.